# Quantifying PON1 on HDL with nanoparticle-gated electrokinetic membrane sensor for accurate cardiovascular risk assessment

Sonu Kumar [1], Nalin Maniya[1], Ceming Wang [1], Satyajyoti Senapati [1] ✉ & Hsueh-Chia Chang [1] ✉

Cardiovascular disease-related deaths (one-third of global deaths) can be reduced with a simple screening test for better biomarkers than the current lipid and lipoprotein profiles. We propose using a highly atheroprotective subset of HDL with colocalized PON1 (PON1-HDL) for superior cardiovascular risk assessment. However, direct quantification of HDL proteomic subclasses are complicated by the peroxides/antioxidants associated with HDL interfering with redox reactions in enzymatic calorimetric and electrochemical immunoassays. Hence, we developed an enzyme-free Nanoparticle-Gated Electrokinetic Membrane Sensor (NGEMS) platform for quantification of PON1-HDL in plasma within 60 min, with a sub-picomolar limit of detection, 3–4 log dynamic range and without needing sample pretreatment or individual-sample calibration. Using NGEMS, we report our study on human plasma PON1-HDL as a cardiovascular risk marker with AUC~0.99 significantly outperforming others (AUC~0.6–0.8), including cholesterol/triglycerides tests. Validation for a larger cohort can establish PON1-HDL as a biomarker that can potentially reshape cardiovascular landscape.

Cardiovascular diseases (CVD) are the leading cause of death globally, with an estimated eighteen million deaths each year[1–4]. CVD generally involves narrowing of the arteries due to cholesterol plaque buildup and blockage of blood vessels through atherosclerosis that prevents the pumping of oxygen and nutrients to vital areas of the body leading to sudden death. The most widely used biomarker for evaluating cardiovascular health is high-density lipoprotein cholesterol (HDL-C) and low-density lipoprotein cholesterol (LDL-C), also known as the good and the bad cholesterol. These two biomarkers are known to have poor sensitivity (~0.6), specificity (~0.6), and Area under the ROC curve (AUC) (<0.7)[5–8], often leading to unsatisfactory CVD diagnosis. Even though the deaths due to CVD can be significantly reduced by simple lifestyle and diet changes, the use of such biomarkers for CVD diagnosis can lead to a false sense of security among the at-risk population. Other markers have been proposed, such as Apolipoprotein AI (ApoAI)[6,9], Apolipoprotein B (Apo B)[5,6], HDL particle (HDL-P)[10,11], and LDL particle (LDL-P)[12–17] but their AUC is still between 0.6 and 0.8[18].

Several reports have shown that a significant portion of the cardioprotective properties of HDL comes from the Paraoxonase 1 (PON1)-containing HDL[8,19–23]. It minimizes LDL oxidation and oxidative stress[22,24] as an antioxidant and reduces the accumulation of cholesterol plague into the macrophages through enhanced reverse cholesterol transport (RCT)[25,26] with natural binding sites on macrophages for PON1-HDL[26]. However, free-floating PON1 does not exhibit atheroprotective property to such an extent[22,27]. Hence, the quantification of only the PON1-HDL in plasma or serum is highly relevant for accurate diagnosis of CVD[28,29]. Currently, there are no immunoassays to quantify the PON1 level on HDL. The presence of lipoprotein-associated lipid peroxides and antioxidants interferes with the enzymatic reaction (HRP)[30–33], making the enzymatic immunoassays unreliable. We have developed elaborate ELISA workflows as a benchmark in this work that can overcome these issues, however, their assay time exceeds 24 h. There are some reports on the quantification of PON1-HDL after isolating HDL from plasma or serum[22,23] using ultracentrifugation (UC)-

[1]Department of Chemical and Biomolecular Engineering, University of Notre Dame, Indiana, USA. ✉e-mail: ssenapat@nd.edu; hchang@nd.edu

based techniques. The use of UC renders the assay slow, tedious, and not scalable for screening of large population. Moreover, upstream isolation also introduces bias due to variable yield of separation[34] methods and dissociation/rearrangement of HDL proteins[35]. Other methods include an inactivation assay that measures the slow inactivation of tightly bound PON1 to HDL but suffers from reproducibility[36]. Hence, there is a need for a phenotype-independent and enzyme-free method to quantify PON1-HDL that does not require any upstream isolation or sample treatment to quantify PON1-HDL, especially if it is to translate to clinical applications.

In this work, we developed a rapid, simple, robust, and portable detection platform for quantification of both PON1-HDL and HDL-P in plasma and serum samples by extending an anion exchange membrane (AEM)-based sensor platform, previously developed to analyze nucleic acids and proteins in plasma and cell culture media[37–39]. The ion-selective property of the AEM allows only counter ions (anion) entry. Upon application of an electric field, the resulting single-direction ion flux creates ion depletion on one side of the membrane surface and ion enrichment on the other side (Fig. 1a). This concentration polarization phenomenon is followed by a negative charge surface polarization phenomenon at a higher voltage which triggers an interfacial electroconvective vortex instability[40,41]. The ion depletion and vortex mixing change the ion current conductance and produce a current-voltage curve (CVC) with distinct underlimiting, limiting, and overlimiting regimes, each with a different differential conductance (Fig. 1b, c). When negatively charged analytes (like DNA or microRNA) bind to the specific probes covalently linked to the depletion side of the AEM, the electroconvective instability is suppressed as these charges are immobile. As a result, the voltage responsible for the overlimiting current in the CVC is shifted by several volts, while its underlimiting region remains unaltered[37,38]. The extent of the shift in the overlimiting regime is directly correlated to the number of targets hybridized to the sensor surface and, after accounting for probe affinity/mass transfer, the target concentration in the bulk sample. Because ion depletion amplifies the induced potential of the charged hybridized species, the voltage shift is much (10–100×) higher than that observed in electrochemical or field-effect transistor sensors. Additionally, the ion depletion and controlled wash remove assay inhibitors and control the ionic strength near the sensing surface, thus making the sensing signal independent of the ionic strength, pH, and chemical composition of the original sample. This feature removes the need to conduct individual-sample calibration and a universal

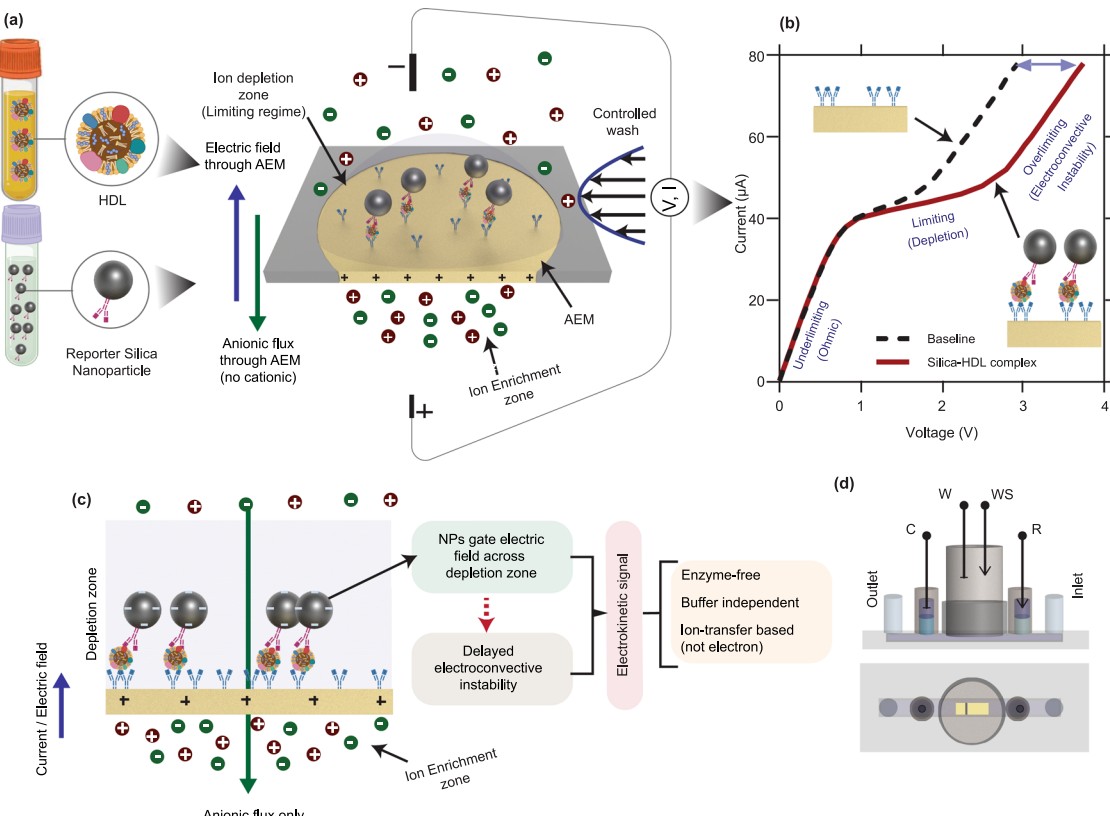

**Fig. 1 | Schematic showing the overview of the Nanoparticle-gated Electrokinetic Membrane Sensor (NGEMS) platform. a** The positively charged anion exchange membrane (AEM) only allows anions to pass, with capture antibodies covalently linked to its surface. A current/electric field is applied across the membrane such that ion depletion zone is formed over the capture surface in the limiting regime. The controlled wash removes non-specifically bound species. A portion of **a** was created with biorender.com. **b** A general CVC curve showing the shift after addition of sample and silica reporters that allows target quantification. **c** Mechanism for the electrokinetic signal produced in the NGEMS platform. The charged nanoparticles gate the electric field across the depletion layer leading to a delayed electroconvective instability. This delay shows up as voltage difference in the overlimiting regime of CVC curve. Due to the signal being of electrokinetic nature, it is not affected by HDL-associated peroxides/antioxidants that would generally affect enzymatic immunoassays. Additionally, the depletion zone ensures

identical conditions above the membrane, ensuring that the voltage signal is independent of bulk ionic concentration/composition−in contrast to field effect transistor sensors. Moreover, because the voltage signal is based on transfer of anions across the membrane, it produces a higher signal than electrochemical assays using electron transfer and is also less susceptible to fouling at the electrodes. **d** Schematic of microfluidic chip used in our work. The inlet is connected to a pump to push sample and assay buffers, 0.1× and 4×PBS. The middle reservoir houses a positively charged anion exchange membrane (AEM) that allows only counter ions (anions) to pass through it. Working W and Counter C electrodes apply an electric field that passes from the middle reservoir into the microfluidic channel that creates a depletion zone on the side facing the microfluidic channel. Working Sense WS and Reference R electrodes measure the voltage difference across the membrane.

standard curve can be used for all samples, including plasma and serum for a given set of capture and reporter probes. For the detection of weakly charged species (like HDL), negatively charged silica nanoparticles attached to a detection antibody were used as reporters in a sandwich design[37,42]. The particle thus brings the necessary negative charge for the sensing signal to the AEM surface upon binding to specific proteins (PON1 or ApoA1) on the HDL (Fig. 1b).

There are additional advantages relevant to the PON1-HDL assay[30,31]. NGEMS does not require any enzymatic reaction to amplify the sensing signal and is hence compatible with lipoproteins and antioxidants, and other bioassay interfering entities. The NGEMS signal comes from the charge on the silica particles and not PON1 or HDL itself; therefore, phenotype effect on the activity of substrates in activity-based assays[36,43] is absent. High charge density on AEM minimizes protein fouling by hydrophobic interaction. Shear from the optimized wash protocol and the electroconvective instability removes non-specifically bound species. Thus, the NGEMS platform does not require any upstream sample pre-treatment step for complex biosamples like plasma and serum. Its inexpensive reagents and simple protocol that can be automated also render it suitable for low-resource settings. Due to the sub-mm dimension of the AEM sensors, PON1-HDL and HDL-P concentration results can be obtained from a single assay in less than 60 min, compared to >24 h for an elaborate ELISA assay we developed for benchmarking. We are not aware of any literature data on the efficacy of human plasma PON1-HDL (different from total PON1) in assessing cardiovascular risk directly.

In this work, we performed a blind pilot study on the efficacy of PON1-HDL in detecting coronary artery disease with 20 clinical samples (10 each in Control and Coronary Artery Disease group) using NGEMS. The study shows that CAD samples contain a significantly lower level of PON1-HDL particles than the control group (AUC 0.99), whereas significantly lower AUC (-0.65) are observed in the same cohort for other biomarkers (HDL-P, HDL/non-HDL cholesterol levels, triglyceride levels, total ApoAI, total PON1, PON1 enzymatic activity, total ApoB, and Oxidized LDL levels). The free-floating PON1 simulated using recombinant version does not seem to produce any significant signal, making NGEMS ideal for identifying proteomic subsets of HDL such as PON1-HDL, PON1-free HDL, or total HDL.

## Results

### The sensing platform and sensing strategy

Figure 1d shows the overall architecture of the NGEMS platform consisting of a microfluidic channel and a few openings for inlet, outlet, electrodes, and a slot to house an AEM sensor at the center of the channel. To record the sensing signal, a current–voltage ($I$–$V$) sweep is applied between the source electrodes (C and W) and reference reservoirs (WS and R). For different current ranges, the three distinct regimes are seen in the current–voltage curve (CVC) of the AEM sensor (Fig. 1). A schematic CVC signal is shown in Fig. 1b, where the introduction of a charged species causes a voltage shift in the overlimiting part of the CVC. Since Apolipoprotein AI is the primary protein on HDL forming over 70% of its overall protein mass[44] and has around 3–7 ApoAI on HDL[45,46]; therefore, to quantify PON1-HDL, AEM surface is functionalized with monoclonal anti-ApoAI as the capture antibody. First, the sample is pushed into the biochip and incubated for 20 min followed by a controlled wash with a high ionic concentration buffer (4×PBS) for 15 s at a flow rate of 0.75 ml/min and then with low ionic concentration CVC measurement buffer (0.1×PBS) at the same flow rate till the CVC shifts between subsequent washes converge. CVC of the AEM is measured and used as a baseline signal. The silica nanoparticle reporter solution in 1×PBS consisting of anti-PON1 (Si-NP) is then incubated for another 20 min, followed by the same wash protocol of 4×PBS and 0.1×PBS. The CVC signal is again recorded and the presence of a shift in the overlimiting regime from the baseline signal confirms the presence of the PON1-HDL (Fig. 2f). The hydrodynamic

shear and high ionic concentration buffer (4×PBS) of the wash protocol remove non-specifically bound charged non-targets and silica particles from the AEM surface by Debye screening. For the detection of PON1-free HDL, silica nanoparticle functionalized with anti-ApoAI (Si-NP') is introduced in the biochip to allow the binding of Si-NP' to the unoccupied HDL which is the PON1-free HDL (Fig. 2c). This produces a second shift for PON1-free HDL (Fig. 2d, f). The Si-NP and Si-NP' are about 5 to 10 times larger than an HDL particle in size; thus, at most, only one silica particle can bind to an HDL particle on the AEM surface. Using this two-step sequential reporter addition strategy, we can also determine the total HDL concentration (HDL-P) by summing the concentrations of PON1-HDL and PON1-free HDL together which has been used throughout this work. Alternatively, we can also determine it directly using a one-step reporter addition strategy (see Fig. 2a, b, e).

### Calibration of NGEMS for detection of PON1-HDL, PON1-free HDL, or the total HDL

The bottlenecks associated with HDL detection in an enzymatic or non-enzymatic assay are related to the bias in the counting of the HDL due to higher affinity for HDL with more ApoAI and reporting the target HDL with the same number of reporters. We overcome the former by operating in a mass-transfer limited regime to remove the affinity bias in the kinetically limited regime. The latter is overcome by using a reporter larger than HDL that only allows one reporter per HDL due to its larger size. A total of 50 nm size of silica is preferred because it is smaller than the Debye layer of DI water, thus allowing it to efficiently gate the electric field with its charge. Smaller reporters like oligoprobes are avoided because different number of reporter can bind to different HDLs.

We constructed the calibration plots for PON1-HDL, PON1-free HDL, and total HDL particles using different known concentrations of HDL. These calibration curves allow us to establish one-to-one correspondence between voltage and bulk concentration, and also serve as a way to account for differences in silica surface charge caused by the different reporter probes on silica. We observe a limit of detection (LOD) of 1 pM for PON1-HDL, as shown in Fig. 2h. The LOD is obtained by first obtaining the limit of blank (LOB), where the experiment is performed using a blank sample (1×PBS), and LOD is taken as three standard deviations higher than the mean providing LOD of 150 mV approximately (see Supplementary Fig. 1). The signal saturates at around 1 nM PON1-HDL concentration, thus providing over a three-decade dynamic range. Similarly, we constructed a calibration plot for PON1-free HDL and total HDL particles (Fig. 2i, j). Both show a LOD of 0.5 pM with a dynamic range of four decades, significantly better than a typical indirect ELISA for total proteins[47]. The total assay time is under 60 min for PON1-HDL and under 90 min for PON1-free and total HDL. Total HDL can be obtained faster by the one-step reporter addition strategy in under 60 min. Steric hindrance by the finite-size silica nanoparticles can contribute to the saturation of the signals, particularly for a multi-step titration assay. It is hence important to operate away from saturation. All our experiments are done at 10× lower than the saturation concentration.

To check whether the CVC shifts are indeed due to the docking of the target and the formation of Ab–Ag–Ab–Si adduct, we used fluorescently labeled silica reporter and captured the confocal image of the AEM sensor with different target concentrations. Figure 3a–e shows a gradual increase in the number of labeled silica particles on the AEM with the increase in PON1-HDL concentration from 1 pM to 10 nM. To further confirm that the voltage shift is not due to free-floating PON1, we created a protein cocktail consisting of rePON1 (recombinant PON1), reApoAI (recombinant ApoAI), and human serum albumin (HSA) in 1:40:1000 molar proportions (estimated ratio in human plasma) with 10 nM rePON1 that corresponds the saturation concentration of PON1-HDL for the AEM sensor (Fig. 2h). As shown in Fig. 3f, only a few fluorescently labeled Si-NP were seen on the AEM

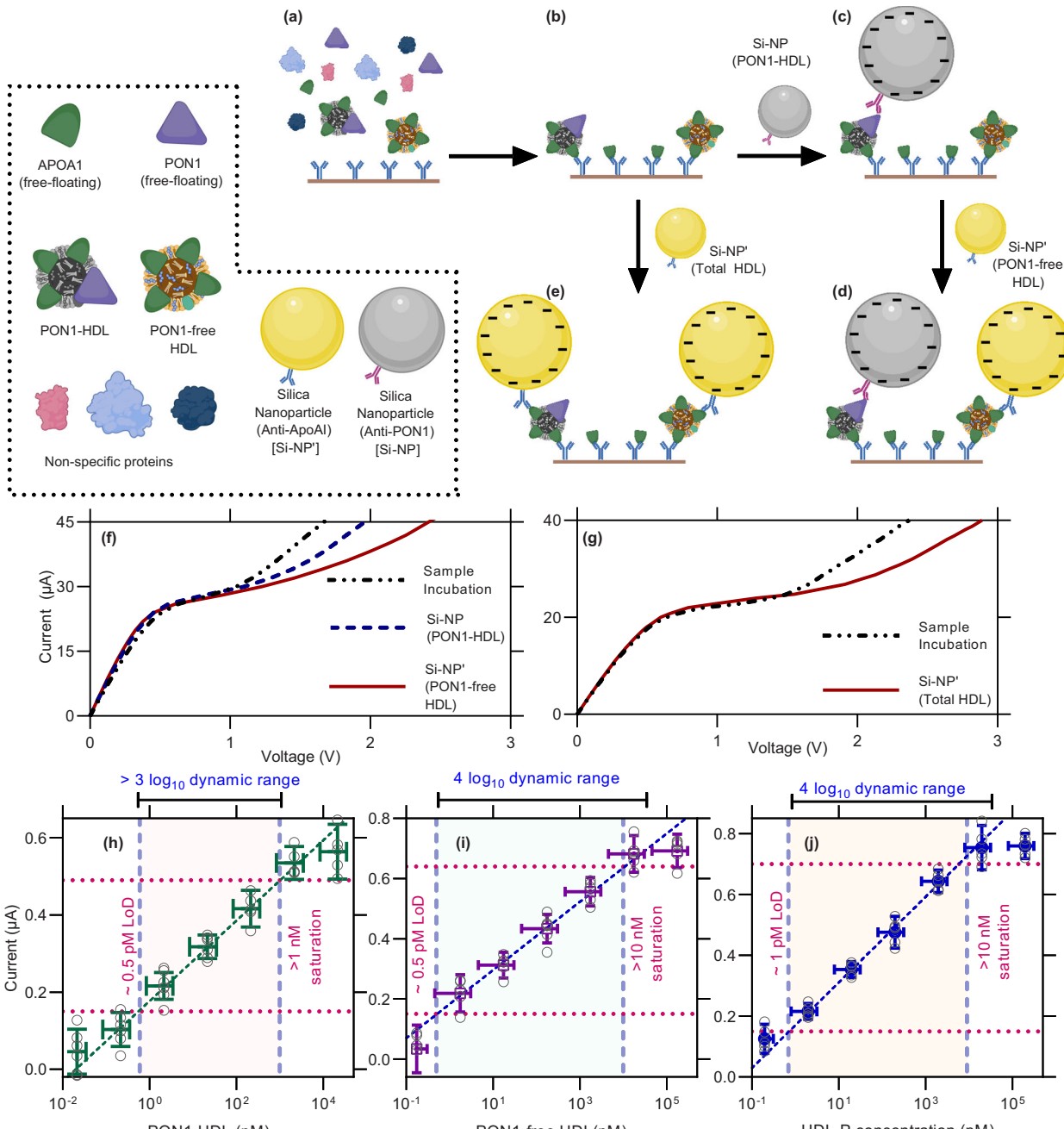

**Fig. 2 | Calibration of voltage shifts to target concentration. a–e** Schematic showing the assay steps of the NGEMS platform. The washing step with 4×PBS and 0.1×PBS buffer is employed after each sample and nanoparticle incubation step before an CVC is recorded. **a** Sample is incubated for 20 min on the anti-ApoAI functionalized AEM surface and then the wash steps employed. **b** Captured HDL on the AEM surface that gives a CVC used as the baseline signal. **c** Addition of silica particles functionalized with anti-PON1 (Si-NP) gives a voltage shift depending on number of captured PON1-HDL. **d** Addition of silica particles with anti-ApoAI (Si-NP') binds to unoccupied HDL in **c** giving a shift proportional to PON1-free HDL. **e** Direct addition of silica reporters with anti-ApoAI (Si-NP') binds to all the HDL giving a shift proportional to HDL-P (total HDL). Silica is significantly larger than HDL thus allowing only one Silica nanoparticle per HDL. **a–e** and its key was created with biorender.com. **f** Typical CVC of two-step sequential (Si-NP and Si-NP') reporter addition strategy as shown in **c** and **d** for PON1-HDL and PON1-free HDL. **g** Typical CVC of one-step Si-NP' reporter addition strategy as shown in **e** for HDL-P (total HDL). Calibration plots for PON1-HDL **h**, PON1-free HDL **i** and HDL-P **j** shown as average of at least five replicates of each concentration with the error bars as the standard deviation. Same sample was measured repeatedly at every given concentration on different NGEMS sensors.

surface compared to its PON1-HDL counterpart for 10 nM (Fig. 3e). The anti-ApoAI probes on the AEM surface cannot capture free-floating PON1 simulated using rePON1, thus preventing the formation of immunosandwich. Additionally, the use of a wrong capture antibody, anti-ApoB, did not show any significant number of fluorescent Si-NPs on the sensor surface (Fig. 3g), thus further confirming the detection selectivity of the NGEMS platform.

Similarly, the confocal images of the membrane with fluorescently labeled Si-NP' for total HDL (Fig. 2a, b, e) are shown for 1 pM, 10 pM, 100 pM, 1 nM and 10 nM in Fig. 3h–l respectively, showing a gradual increase in fluorescent Si-NP' with HDL-P for PON1-HDL. To ensure that only HDL particles are detected and not free-floating ApoAI that could also be present in samples, the same monoclonal antibody on the silica reporter is used that we also used on the AEM surface. This ensures

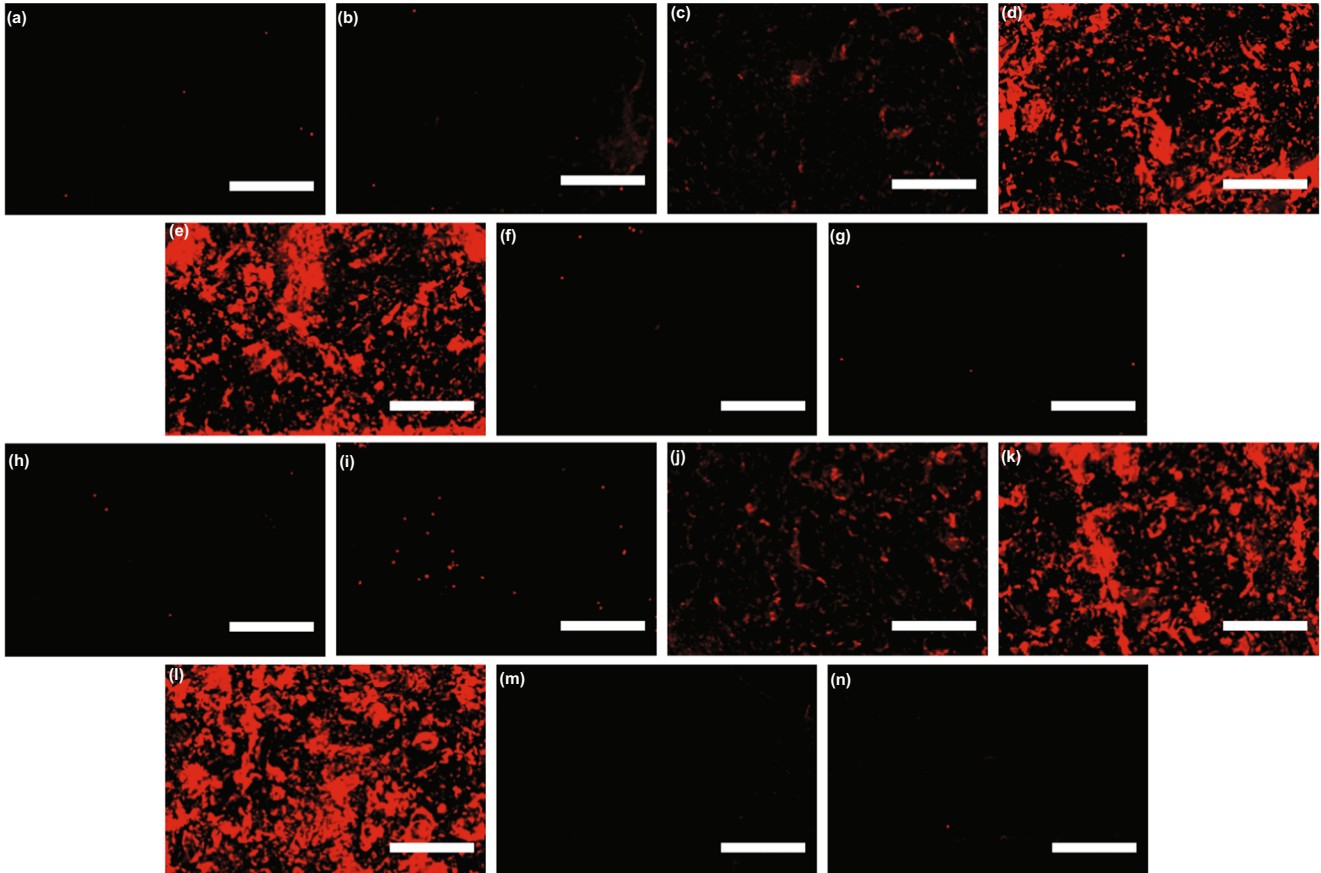

**Fig. 3 | Confocal images using red fluorescent silica nanoparticles. a–g** Use fluorescent Si-NP and **h–n** uses fluorescent Si-NP' both added directly after sample incubation. All the images of membranes had anti-ApoAI as the capture antibody except for **g** and **n** that had Anti-ApoB. The images show the state of membrane for sample containing **a** 1 pM, **b** 10 pM, **c** 100 pM, **d** 1 nM and **e** 10 nM of PON1-HDL and **h** 1 pM, **i** 10 pM, **j** 100pM, **k** 1 nM and **l** 10 nM of HDL-P. A protein cocktail containing rePON1, reApoAI and HSA (1:40:1000) that contained an equivalent protein in 10 nM PON1-HDL and total HDL respectively, i.e., **f** 10 nM rePON1 + 400 nM reApoAI + 10000 nM HSA and **m** 1 nM rePON1 + 40 nM reApoAI + 1000 nM HSA were used as controls. **g** and **n** used 10 nM of HDL as sample but with anti-ApoB, thus providing no signal. Scale bars are 100$\mu$m. Imaging was done once for each case.

that immunosandwich will not form with a single ApoAI-containing species as the epitope of captured ApoAI will be occupied by the capture anti-ApoAI. Figure 3m shows no fluorescently labeled Si-NP' of the immunosandwich on AEM when a protein cocktail of rePON1, reApoAI, and human serum albumin (HSA) in 1:40:1000 molar proportions with 40 nM reApoAI (roughly equivalent to ApoAI in 10 nM HDL-P) is used. Additionally, the use of a non-target capture antibody such as anti-ApoB also does not show significant fluorescent Si-NP' on the membrane (Fig. 3n). Hence, these confocal findings conclude that free-floating proteins do not produce any signal on the NGEMS platform, demonstrating the high specificity of the platform. Moreover, all three calibration curves are statistically identical, despite different detection antibodies, indicating unbiased reporting of HDL by the silica nanoparticle reporters (Fig. 2h–j).

**Mass-transfer limited regime: short incubation time and robustness**

We have measured several concentrations of PON1-HDL and HDL-P at twenty minutes and one hour incubation times at the low and high concentrations and conclude that 20 min is enough to reach a steady state signal as shown in Fig. 4a–d. The reason we require only 20 min for a steady signal is because we operate in the mass-transfer limited regime and never reach the true adsorption equilibrium. For a small sensor of size $a$ (~100 $\mu$m), depletion of analyte within a depletion volume above the membrane of radius $a$ occurs within a time of $a^2/D$ where $D$ is the diffusivity of the analyte if the on-rate of

antigen–antibody interaction is high enough[48,49]. After this time, the flux is significantly lower and the analyte concentration above the sensor asymptotically reaches the true adsorption equilibrium exponentially over a very long time. This is verified by changing the surface antibody concentration to one-twentieth of its original value and at a concentration close to saturation. Because the adsorption kinetics depends on the surface concentration of the antibody, but mass-transfer does not, the signal only changes with lower probe concentration if our platform is in kinetically limited regime. We observe that the signal remains unchanged at the higher end of the dynamic range (Fig. 4e). This confirms we are in the mass-transfer limited regime. With a small sensor, we only capture analyte within 10–100 nL, thus, the sensor needs to be very sensitive to detect the small number of captured analytes. This is possible for our sensor because of silica nanoparticle and the ion-depletion action of the membrane amplify the voltage signal and may not be possible for other sensors operating in this mass-transfer limiting regime. Working in the mass-transfer limited region also means that the platform is more robust—the signal is less sensitive to interfering agents that reduces the affinity of the antibody. Individual-sample calibration becomes unnecessary due to this insensitivity to the binding kinetics.

We have also conducted some numerical simulation of this rapid depletion (Fig. 4f–i). As can be seen, majority of the depletion occurs within the first few minutes, and the surface concentration of the analyte away from the depletion volume above the membrane does not change over time significantly. The concentration profile directly

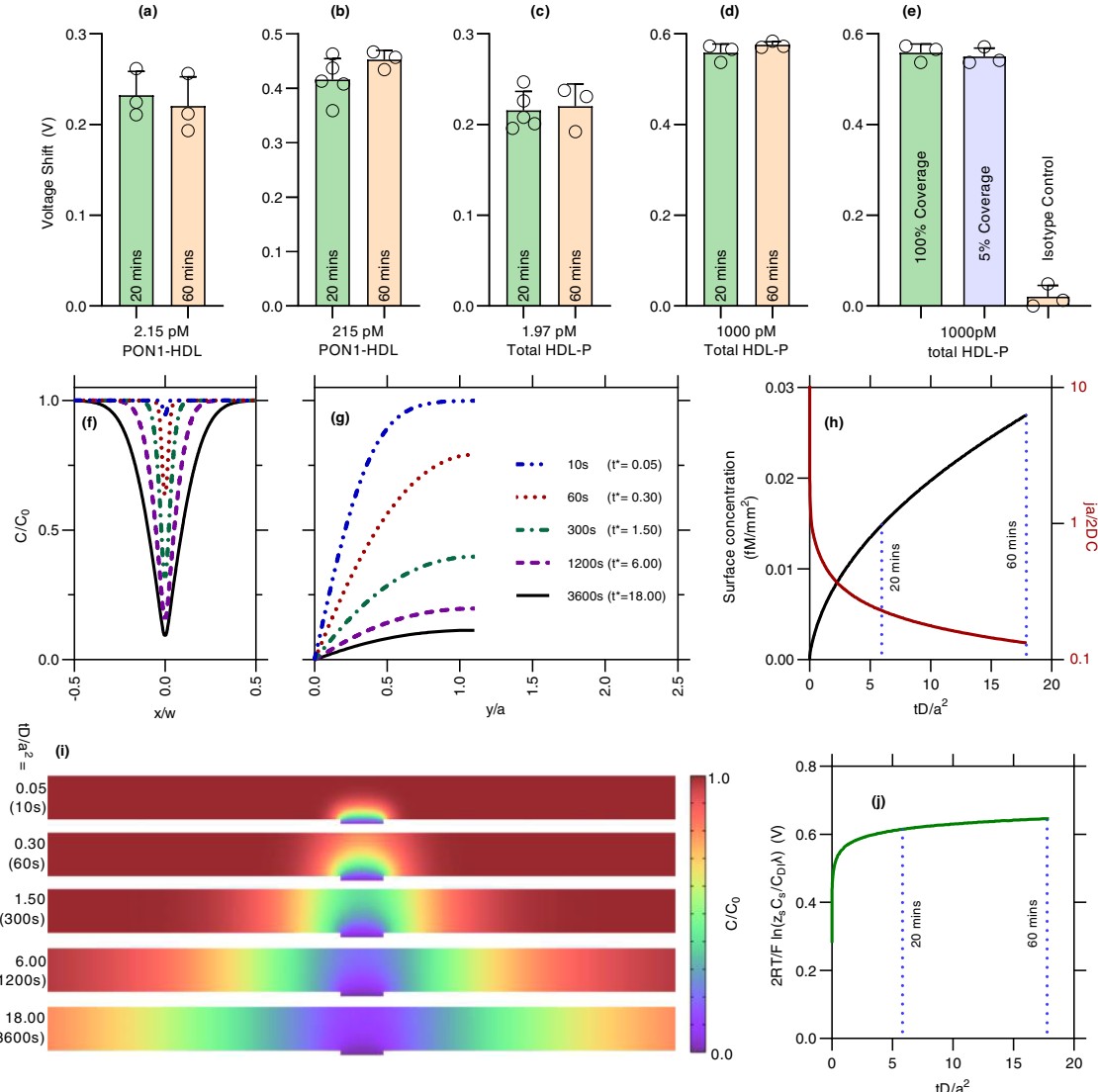

**Fig. 4 | Variation of signal with time and effect of mass-transfer limitation for a small sensor.** **a**–**d** shows the signal at different times for 2.15 pM PON1-HDL, 215 pM PON1-HDL, 1.97 pM HDL-P, and 1000 pM HDL-P showing almost identical signals between 20 and 60 min. **e** Effect of antibody surface coverage on the overall signal with 100%, 5% and 0% coverage. Similar signal between 100% and 5% suggests we are in mass-transfer limited regime. **f**, **g** shows the concentration at the channel center along the channel and directly above the membrane (membrane located at origin), and how the concentration gradient decreases over time. Shared figure legend for both **f** and **g** shown in the latter. **h** Surface concentration of analyte on membrane surface and non-dimensional flux with time. **i** Zoomed-in snapshots of the microfluidic channel from numerical simulations in mass-transfer limited regime (irreversible reaction on the membrane surface). **j** Theoretical signal increase over time showing a pseudo-steady state after t* = 1. Each case in **a**–**e** was measured three times independently except for 20 min case of **b**, **c** where they were independently measured five times with error bars as one standard deviation. Same sample was measured repeatedly at every given concentration on different NGEMS sensors.

above the membrane and at the center of the channel are also shown. From our work on how surface charge affects the voltage shift[42], the voltage signal is $2RT/F \ln(z_s C_s/c_0\lambda)$ where $C_s$ is the surface concentration of the analyte on the surface, $z_s$ is the charge on silica nanoreporters, $c_0$ is ionic concentration of water ($10^{-7}M$) and $\lambda = 100$ nm is the debye layer at DI conditions. Due to logarithmic dependence on the concentration, this theoretical signal also mostly saturates after $a^2/D$ reaching a psuedo-steady state (very slow diffusive flux towards true equilibrium[49]) as seen in Fig. 4j.

### Control experiments through HDL delipidation and pooled human plasma samples

As the voltage signal of the NGEMS for PON1-HDL, PON1-free HDL and total HDL comes from the negatively charged silica nanoparticle reporter attached to the HDL, the signal should then disappear if we dissolve the HDL particle using a detergent solution (0.05% Tween 20, 1% BSA in 1×PBS). As shown in Fig. 5a, the CVC returns close to the baseline when sequentially treated AEM sensor with SI-NP and SI-NP' was exposed to the detergent solution. Further sequential addition of Si-NP and Si-NP' should not produce any shift as there is no HDL on the AEM surface, which is confirmed in Fig. 5a. However, the delipidation leaves the ApoAI-anti ApoAI adduct on the AEM, where anti-ApoAI antibody is covalently attached to the AEM surface. Hence, adding Si-NP'' (polyclonal anti-ApoAI) produces a shift as the particle can now bind to a different epitope on ApoAI-anti ApoAI adduct. We also performed the confocal imaging of delipidation step. Figure 5b, c shows significantly fewer fluorescent silica reporters on the AEM surface after delipidation. Furthermore, the addition of pre-delipidated HDL (diluted in detergent solution) and fluorescently labeled Si-NP/Si-NP' also shows a negligible fluorescence signal on the AEM surface (Fig. 5d, e)

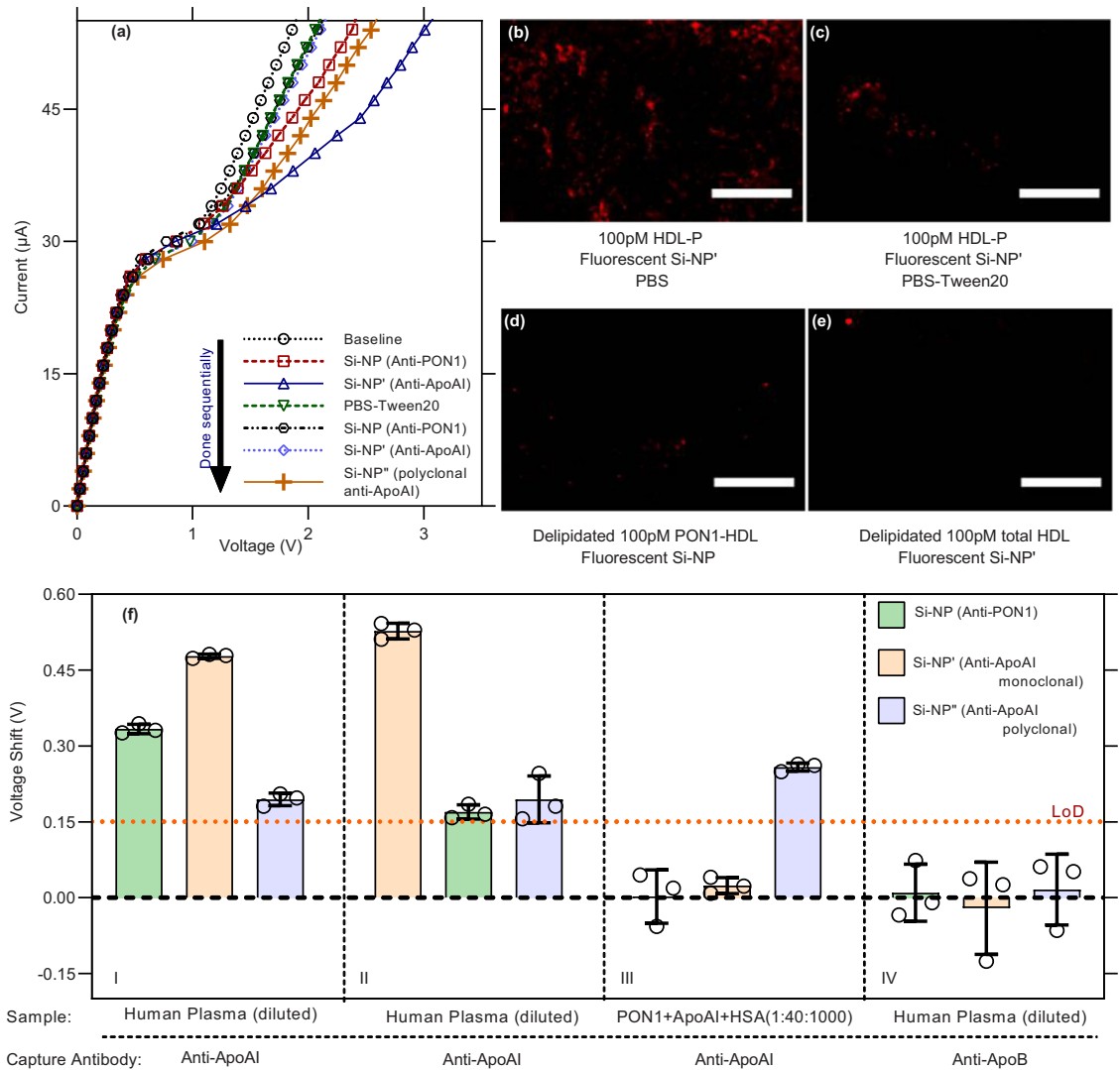

**Fig. 5 | Control experiments using NGEMS. a** Effect of delipidation on the CVC. Delipidation brings back the shifted CVC close to its baseline. Further addition of Si-NP/Si-NP' does not cause any shift but an addition of Si-NP'' (silica attached to polyclonal anti-ApoAI) does cause a shift because of the presence of lipid-free ApoAI on the surface. Confocal image of the AEM surface **b** when treated with 100 pM HDL followed by addition of fluorescent Si-NP' and treated with PBS for 30 min as the control, **c** when treated with 100 pM HDL-P followed by addition of fluorescent Si-NP' and treated with PBS-Tween20 for 30 min for delipidation, **d** when predelipidated 100 pM PON1-HDL is used followed by addition of fluorescent Si-NP, and **e** when predelipidated 100 pM HDL-P is used followed by addition of fluorescent Si-NP'. For both **d**, **e**, detergent treatment is done at higher HDL concentration and then diluted with PBS to achieve desired concentration. **f** Control experiments with 50000× diluted pooled human and plasma protein cocktails at different orders of Si-NPs, and different capture antibodies and samples. Error bars represent one standard deviation and each case in **f** are triplicated with same sample measured repeatedly on different NGEMS sensors. Scale bars are 100 μm for **b–e** and were only repeated once.

demonstrating only intact HDL can produce a fluorescence signal confirming the high detection selectivity of the NGEMS platform.

Additional control experiments with diluted pooled human plasma are presented in Fig. 5f and are divided into four subgroups (I–IV). In the first subgroup, an experiment is conducted detection of PON1-HDL and PON1-free HDL by sequential addition of Si-NP and Si-NP' reporters, but at the end of the experiment, Si-NP'' reporters are added. This produced a signal from free-floating ApoAI (see I of Fig. 5f). Next, we changed the order of silica reporter addition and first added the Si-NP' reporter. In that case, as shown in II of Fig. 5f, the first shift is synonymous with the one-step scheme for quantification of total HDL concentration (Fig. 2a, b, e). The addition of Si-NP after Si-NP' does not produce a shift due to one silica per HDL condition and all HDL being occupied by Si-NP' in the first step. Si-NP'', on the other hand, can still produce a shift from the free-floating ApoAI captured on the surface. With a protein cocktail with a similar concentration of total PON1,

ApoAI and Albumin as in the plasma (to mimic free-floating proteins), no signal is seen, as shown in III of Fig. 5f with Si-NP and Si-NP'. With the addition of Si-NP'', a signal will still be produced with reApoAI simulating free-floating ApoAI in the cocktail due to capture and binding of reporter to different epitopes. Lastly (see subgroup IV of Fig. 5f), if a non-target antibody anti-ApoB was used as a capture antibody, none of the Si-NP, Si-NP', and Si-NP'' produce any shift as no target species was captured on the surface.

## Benchmarking of NGEMS against other techniques using human plasma

As there are currently no other assays for PON1-HDL to benchmark against NGEMS, we developed a new ELISA scheme, ELISA-1 (described in detail in the methods section). Briefly, both free-floating PON1 and PON1-HDL in plasma samples are first allowed to bind with anti-PON1 antibodies attached to an ELISA microwell. Then polyclonal anti-PON1

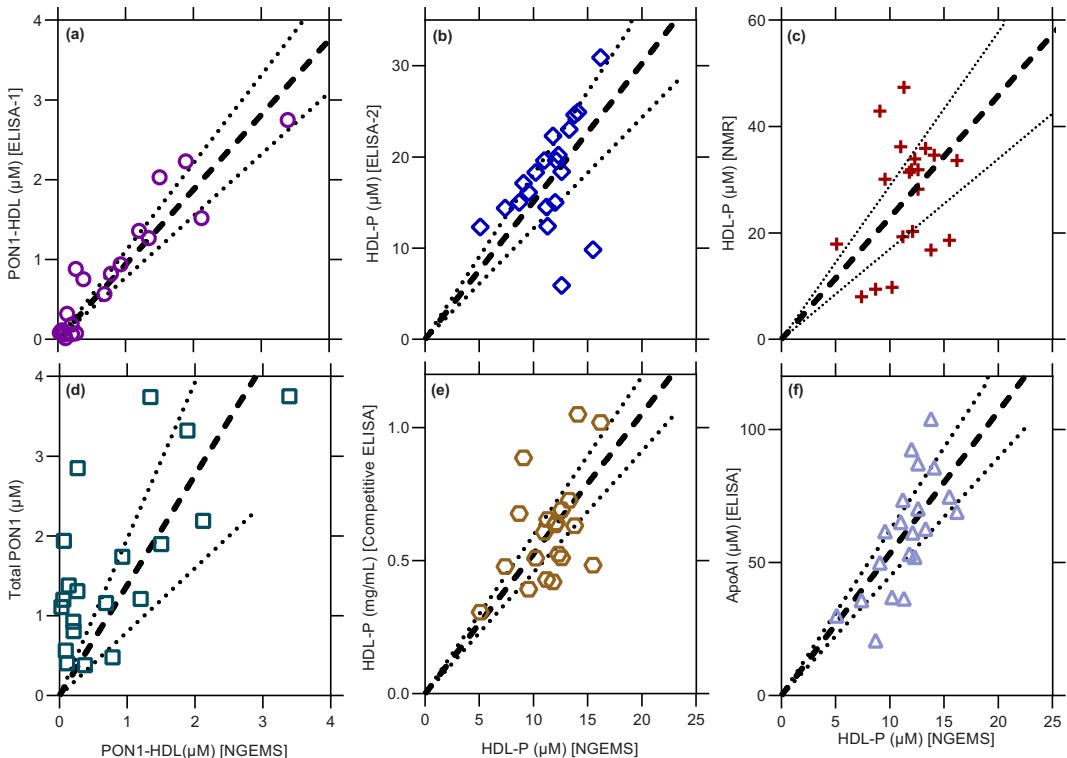

**Fig. 6 | Correlation between different measurements from independent platforms. a** PON1-HDL from NGEMS vs from ELISA-1. **b** HDL-P from NGEMS vs ELISA-2. **c** HDL-P from NGEMS vs 1H-NMR. **d** Variation of PON1-HDL versus the total PON1. **e** HDL-P from NGEMS vs competitive ELISA. **f** Variation of HDL-P versus total ApoAI.

All the points are measured in triplicates from 20 independent human plasma samples. Thick dashed line represents the best fit with zero intercept for each case where the dotted line represents the 99% confidence band.

with inactivated HRP is introduced that selectively binds with the free-floating PON1 as the PON1 on PON1-HDL is sandwiched between HDL and capture surface (ELISA microwell) and hence PON1-HDL is not accessible. The HDL is then delipidated, which makes the PON1 on HDL accessible and is allowed to bind with HRP-conjugated polyclonal anti-PON1, thus producing a signal for PON1-HDL (see methods section for detailed protocol and Supplementary Fig. 2a in the supplementary information). As shown in Fig. 6a, both NGEMS and ELISA-1 techniques correlate well and show a slope of almost one. This also shows that NGEMS does not need to be calibrated for a given set of probes due to the highly robust electrokinetic signal that is, unlike ELISA-based techniques, not affected by environmental noise and variations. Interestingly, PON1-HDL does not correlate well with total PON1 due to the presence of free-floating PON1 in the plasma as shown in Fig. 6d. It must also be noted that ELISA-1 took over 24 h for quantification of PON1-HDL, while the NGEMS platform required only less than an hour. For benchmarking of HDL-P, we use 1H-NMR (a gold standard technique for HDL-P but known to overpredict)[45], a commercially available competitive ELISA scheme, and ELISA-2, which uses a competitive capture strategy during HDL delipidation to capture one ApoAI per HDL (see methods and Supplementary Fig. 2b). We see a good correlation between HDL-P (total HDL) from NGEMS and other techniques as shown in Fig. 6b, c, e.

Additionally, we use the unique structural property that the HDL contain 2–8 ApoAI per HDL to evaluate our platform by plotting total ApoAI against HDL-P from our platform in Fig. 6f, which is a standard evaluation test for HDL platforms. A slope of 3-5 is expected if the platform counts HDL-P correctly. HDL-P from NGEMS predicts roughly 4–6 ApoAI per HDL as shown in Fig. 6f, which is very close to the actual reported number of ApoAI per HDL[45,46], and is very difficult to get with NMR or Ion Mobility Analyzer-based methods due to bias towards overprediction and underprediction of HDL-P, respectively[45].

## Blind pilot study with patients and controls of coronary artery disease (CAD) and comparison to other commonly used biomarkers

To test the efficacy of the NGEMS platform for PON1-HDL, PON1-free HDL, and HDL-P, we designed a pilot study with 10 CAD and 10 control group samples. Apart from NGEMS, other platforms routinely used to assess cardiovascular risk were also compared along with our novel ELISA schemes ELISA-1 and ELISA-2 (see methods and Supplementary Fig. 2). Figure 7 summarizes the results from the study, where each subfigure is labeled with the biomarker, assay/technique used, and their corresponding AUC values from the respective ROC plot (Supplementary Fig. 3). The pilot clinical study clearly shows that the PON1-HDL outperformed the other biomarkers with PON1-HDL AUC of 0.99 with NGEMS and 0.90 with ELISA-1. The lower AUC values of ELISA-1 can be attributed to the incomplete inactivation of reporter antibodies used to block free-floating PON1, thus producing an unwanted signal from the enzymatic reaction at those partially inactivated sites. Moreover, ELISA-1 takes >24 h for PON1-HDL compared to 60 min with NGEMS. The total PON1 using a sandwich ELISA got AUC ~0.83, and the activity-based PON1 assay received an AUC ~ 0.67, thus showing that it is the PON1-HDL that has the highest efficacy in an accurate risk assessment and not the measured PON1 activity or the total PON1. Moreover, PON1 activity assays are affected by chelating agents such as EDTA commonly added to plasma for storage and stability[50]. Cholesterol/Triglyceride levels, the most widely used biomarker for cardiovascular risk assessment, performed worse than PON1-HDL in distinguishing the two groups with AUC ~ 0.65. HDL-P was measured using four different techniques—NGEMS, Competitive ELISA, 1H-NMR, and ELISA-2 showing AUC ~ 0.55-0.65 for the first three and 0.775 for ELISA-2. The 1H-NMR method for HDL-P quantification, also the most well-established method for HDL-P quantification, showed a similar AUC to that of NGEMS. The AUC for these already established

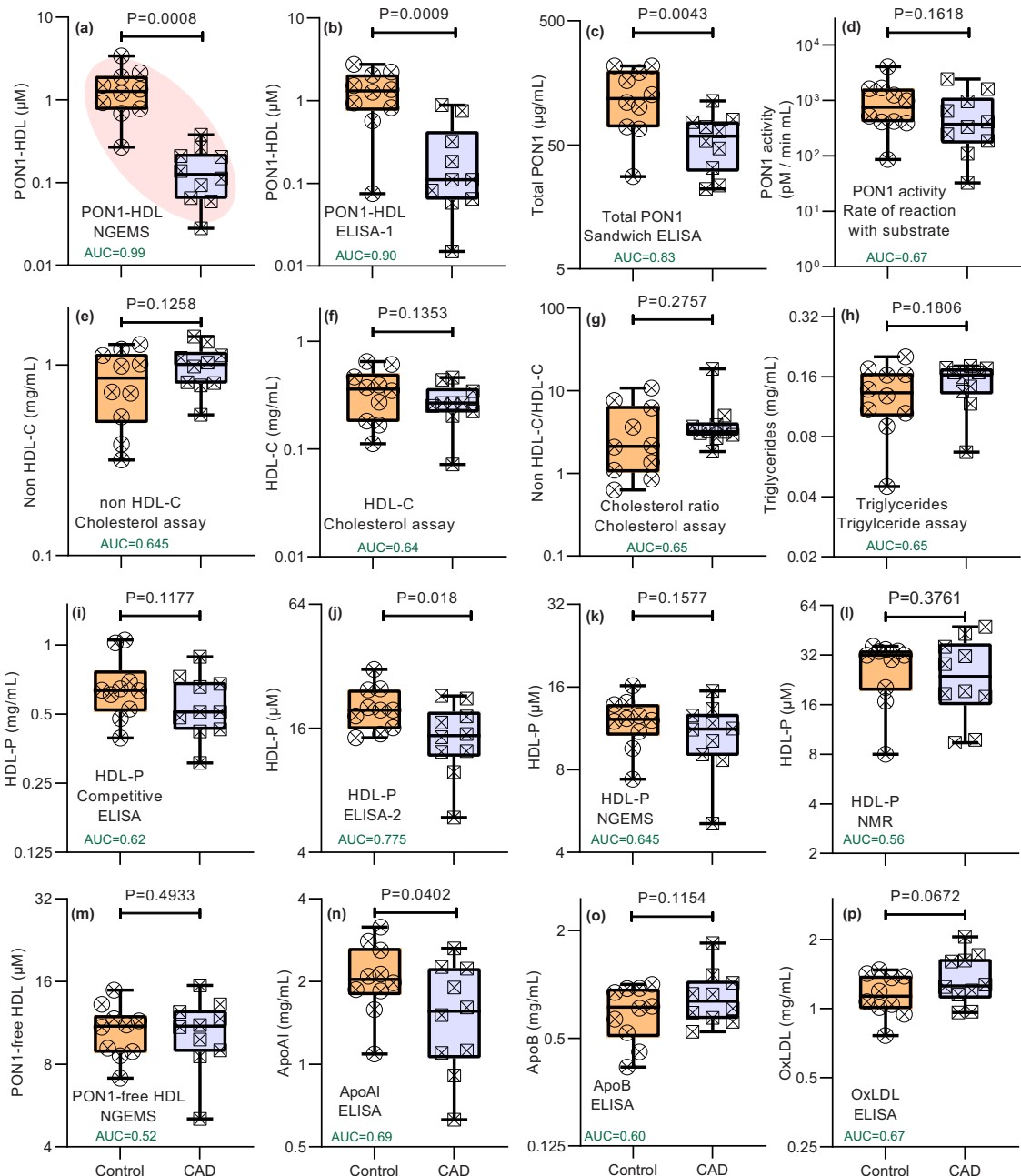

**Fig. 7 | Blind study with clinical samples comparing PON1-HDL to other biomarkers from different platforms. a** PON1-HDL using NGEMS (AUC = 0.99). **b** PON1-HDL using ELISA-1 (AUC = 0.90). **c** Total PON1 using sandwich ELISA (AUC = 0.83). **d** PON1 activity assay using rate of reaction with substrate (AUC = 0.67). **e** Non-HDL cholesterol (AUC = 0.645), **f** HDL cholesterol (AUC = 0.64) and **g** cholesterol ratio (AUC = 0.65) from cholesterol assay. **h** Triglyceride levels from triglyceride assay (AUC = 0.65). Total HDL particle concentration HDL-P from **i** competitive ELISA (AUC = 0.62), **j** ELISA-2 (AUC = 0.775), **k** NGEMS as sum of PON1-HDL and PON1-free HDL (AUC = 0.645) and **l** 1H-NMR (AUC = 0.61). **m** PON1-free HDL from NGEMS (AUC = 0.52). **n** ApoAI from sandwich ELISA (AUC = 0.69). **o** ApoB from sandwich ELISA (AUC = 0.60). **p** OxLDL from commercial ELISA (AUC = 0.67). p-values are calculated from unpaired parametric one-tailed t-test with Welch's correction. AUC values calculated from ROC plots (see Supplementary Fig. 3 for ROC plots). Each datapoint was average of several replicates. Every sample in the figures are measured in triplicates except in **e**–**g** where they are duplicated. Each plot is a standard box and whisker plot with central line being the median, box being the 25th and 75th quantile and whiskers representing 0th and 100th quantile. Both controls and CAD groups had ten samples ($n = 10$) each.

biomarkers matched very well with literature values suggesting the cohort is representative[8,51–57]. Nevertheless, none of these methods for HDL-P approach the AUC values of PON1-HDL from NGEMS. Other biomarkers, including ApoAI, OxLDL, and ApoB also significantly underperform compared to PON1-HDL. Importantly, PON1-HDL varies by two orders in the cohort, with its median between control and CAD groups different by one order of magnitude, which is significantly higher than all other biomarkers. The results for PON1-HDL suggest >95% sensitivity and specificity. On the contrary, PON1-free HDL showed an AUC ~ 0.5, indicating that PON1-free HDL may not have any distinguishing power between the two groups.

## Discussion

A Nanoparticle-Gated Electrokinetic Membrane Sensor (NGEMS) platform is developed to overcome several challenges for the quantification of PON1-HDL, PON1-free HDL, and the total HDL particle concentration (HDL-P). Free-floating PON1 does not produce a signal with NGEMS, allowing us to measure the concentration of PON1-HDL

specifically, resulting in AUC = 0.99. The other biomarkers commonly used to assess the cardiovascular risk, such as HDL-C, non-HDL-C, triglycerides, ApoAI, ApoB, and HDL-P, were compared to PON1-HDL and performed poorly in distinguishing the two groups, with most of them having AUC in the 0.6-0.7 range. Interestingly, PON1-free HDL was found to be incapable of distinguishing the two groups (AUC ~ 0.5). We are currently pursuing a significantly larger cohort of patients with a longitudinal study. With the results validated for a larger pool of patients from independent studies, PON1-HDL could potentially change the cardiovascular landscape and perhaps even correlate it with future risk.

Moreover, due to its microfluidics nature, our platform can be fully automated with peristaltic pumps and potentiostats/galvanostats. A voltage or current sweep across the IV curve which is <4 V and <200 microamperes means that the power requirement of such a setup can be supplied with the help of a small power supply for both the pumps and the potentiostat. A first-generation prototype is currently in development. Turn-key operation, low cost and fast measurement is the end-goal in terms of automation of this platform. After receiving feedback from clinicians for the first prototype, we plan to develop a second-generation prototype to conduct a large-scale clinical study that would demonstrate the usefulness of our platform and the validity of PON1-HDL as a marker. Our platform can also, in theory, identify any other proteomic subclass of HDL in their native form, which cannot be done by any of the commonly used techniques – often relying on surrogate measurement of the interested subclass[58].

## Methods

### Ethics statement
Received coronary artery disease and control group samples from Precision for Medicine, and unlabeled and randomized immediately upon arrival. An approved IRB protocol is already in place at Precision for the collection of plasma samples from patients. Precision for Medicine works with regulatory authorities and accrediting organizations around the world to ensure that the sample collection process and protocol follow the latest FDA, EMA, and MHRA guidelines. The pooled healthy plasma samples were obtained from Innovative Research. All procedures performed in studies involving human participants were in accordance with the ethical standards of the University of Notre Dame.

### Fabrication of biochip
A microfluidic chip with a channel of 3 mm × 30 mm × 0.3 mm was constructed, as shown in Fig. 1. Three layers of polycarbonate sheet of 0.3 mm thickness is used to fabricate the biochip using our reported protocol[35]. The sensor is fabricated by embedding a small piece of anion exchange membrane (*Mega a.s.*, Czech Republic) of dimension 0.3 mm² in an epoxy resin (TAP Quik-Cast, Tap plastic)[35]. The fabricated AEM-based sensing module is then housed at the center of the channel with an inlet, outlet at the two ends, and two reservoirs for a four-electrode potentiometry, as shown in Fig. 1. The working (W) and counter (C) electrode reservoirs were used to pass an electric current across the AEM sensor while the reference (R) and working sense (WS) reservoirs were used to measure the voltage difference across the membrane using a Gamry potentiostat 500. A peristaltic pump is connected to the inlet to pump the sample and assay buffers.

### Antibody functionalization (silica reporter/AEM surface)
The antibodies are functionalized on the ANM surface using the optimized protocol reported earlier[35]. Briefly, the carboxyl groups on the AEM surface were first created by UV treatment of 3,3′,4,4′-benzophenonetetracarboxylic acid solution followed by 2 h of incubation of the AEM at low pH 2–3. The AEM sensor was then treated with 8% EDC (Life Technologies 24510) in MES buffer for 20 min, followed by the addition of 0.1 μg (0.02 mg/mL) anti-ApoAI (Abcam ab52945).

The membrane is hydrophilic; therefore, no further blocking steps are required. For altering the surface coverage of antibodies on the surface to five percent, an isotype control antibody was allowed to compete with the actual target antibody for EDC/NHS crosslinking reaction with only 5% of the total antibody for the target and the rest as isotype control, while maintaining the total amount (isotype + target antibody) the same. Similarly, the reporter particles were prepared by conjugating antibodies with 50 nm carboxylated silica particles using the EDC-NHS coupling chemistry. First, silica particles were suspended in MES buffer (ThermoFisher BupH #28390) and washed thoroughly by centrifuging at a 15000–17000 g and resuspension (by sonication), followed by treatment with 4% EDC and 4% Sulfo-NHS (ThermoFisher #24510) for one hour with continuous mixing to activate the carboxyl group. The excess EDC and Sulfo-NHS were removed by centrifugation and resuspension like before. The silica particles were then incubated overnight with 2 μg anti-PON1 for PON1-HDL (Abcam ab24261) or anti-ApoAI for total HDL (Abcam ab52945) in 1×PBS for covalent linkage of antibodies. In some cases, where polyclonal anti-ApoAI was used during control experiments, same amount of polyclonal anti-ApoAI was added. Fluorescent silica particles (DNG-L083) were also functionalized similarly. All antibodies were conjugated in azide-free environment.

### HDL concentration for calibration
To calibrate voltages against known concentrations of PON1-HDL and HDL-P, commercially available HDL solution was used (Sigma Aldrich SAE0054-10MG). The pure HDL does not contain any free-floating PON1, therefore total PON1 was used as the concentration of PON1-HDL. For HDL-P, ELISA-2 was used to obtain the concentration.

### Cholesterol, triglyceride, and other assays
Cholesterol assay (Abcam ab65390), triglyceride assay (Abcam ab65336), PON1 activity (Abcam ab241044) assay, Oxidized LDL ELISA Kit (Abcam ab242301), ApoB ELISA kit (Fisher Scientific EH34RB) and HDL competitive ELISA kit (Novus Biologicals NBP2-60508) were obtained commercially and performed according to manufacturers' guidelines.

### In-house total protein ELISA kits
The in-house ELISA kits were created by first coating high binding microwell plates with 2–5 μg/mL of anti-ApoAI (Life Technologies MIA1402) for ApoAI ELISA and anti-PON1 (Abcam ab24261) for PON1 ELISA overnight followed by blocking with Casein buffer (Bioworld # 40320020-1) for both total ApoAI and total PON1 ELISA. Commercially obtained recombinant proteins for ApoAI (Novus Biologicals NBP2-34869-100ug) and PON1 (Biovision P1556-10) were used as standards for ELISA. The microwells were always washed with PBS-T (0.1% Tween20 in PBS), and samples were always diluted in assay diluent (2% BSA, 0.05% Tween20 in PBS). The sample was incubated for 12 h overnight, followed by three washes with PBS-Tween solution. Reporter antibodies with Horseradish Peroxidase (HRP) for ApoAI (VWR 10680-872) or PON1 (VWR 10408-588) at manufacturer's suggested dilution were then added and incubated for an hour, followed by a wash and addition of o-Phenylenediamine dihydrochloride (Sigma-Aldrich P9187) with 0.4 mg/mL OPD, 0.4 mg/mL urea hydrogen peroxide in 0.05 M phosphate-citrate buffer as the substrate.

### ELISA-1
This method's assay diluent consists of only 2% BSA in 1xPBS. The microwell is coated with anti-PON1 as mentioned in the previous subsection, and the sample is diluted in 2% BSA-PBS to ensure HDL remains intact while allowing free-floating PON1 and PON1-HDL both to bind to the capture antibodies on the microwell surface. Following sample incubation for 24 h and washing with 1xPBS, PON1 on PON1-HDL is sandwiched between the capture surface and the HDL, and thus its epitopes are inaccessible to any new antibodies we add. Therefore,

we add anti-PON1 (VWR 10408-588, 1:1000 dilution) with inactivated HRP, which allows these antibodies to bind to all the free-floating PON1 captured on the surface and not to PON1 on PON1-HDL. After incubating these antibodies for 2 h, we add 2%BSA with 0.05% Tween 20 in the microwell for 3 h, allowing the HDL to delipidate, followed by several washes with 0.05% Tween 20. Anti-PON1 (VWR 10408-588) with active HRP is then added to the wells, which binds to PON1, which was inaccessible before delipidation for one hour, followed by wash and reaction with substrate. Therefore, the reaction only occurs at the locations where antibodies with active HRP are present, and thus only PON1-HDL produces a signal. For the PON1 standard, the addition of inactivated antibodies is skipped and instead incubated with active HRP antibodies in the end. See Supplementary Fig. 2a for the proposed workflow.

### ELISA-2

The microwell surface is coated with anti-ApoAI (Life Technologies MIA1402) as described for ELISA-1, and detergent-free assay diluent (2% BSA in 1×PBS) is used to dilute the samples to ensure HDL remains intact. Following 24-h incubation, the microwells are washed with 1×PBS, followed by the addition of 2% BSA, 0.05% Tween20, and 20 µg/mL of capture antibody in 1xPBS to the microwell. The ApoAI from HDL bound to the surface remains bound during the delipidation, but the other ApoAI on HDL (-3–7) get solubilized. These ApoAI, once released, will experience competition to bind with the capture antibody on the surface and the same capture antibodies in the solution (added with the delipidation solution). The high concentration of antibodies in the solution outcompetes the surface antibodies allowing only one ApoAI per HDL to remain on the surface, whose concentration can then be determined by the addition of HRP-conjugated anti-ApoAI (VWR 10680-872, 1:1000 dilution), wash steps, and the substrate reaction with OPD. The standard is treated identically. This method produced a signal with free-floating ApoAI but they are generally not present in large quantities even in plasma samples. See Supplementary Fig. 2b for the proposed workflow.

### Confocal imaging

The membranes from the fluorescent silica experiments were mounted on a glass slide with a coverslip as imaged using an Upright Nikon C2 + Laser Scanning Confocal Microscope at 561 nm laser line (Texas red) using z-stacking to take a series of images that are then converted into EDF focused image. The settings are kept the same across all the images taken using a 10× objective. ImageJ is then used to enhance the image brightness and contrast 4× times, and it is made sure that less than ten percent of the pixels are saturated even at the highest concentration.

### 1H-NMR

1H-NMR was performed using a 500 MHz Bruker instrument at the NMR facility at the University of Notre Dame using standard presaturation pulses (*zgpr*). The peak deconvolution was carried out as per previous studies[12,59–66]. Plasma was separated into 15 different subfractions based on its density using Density Gradient Ultracentrifugation as per Chapman et al.[67], and reference NMR methyl peaks were constructed. The plasma peaks were then deconvoluted using these reference peaks using MATLAB and a few digitally shifted references to obtain the methyl peak in different lipoprotein fractions, which were then converted to lipoprotein particle concentration based on the known composition[68].

### Human patient sample

All the clinical plasma samples were obtained commercially from Precision for Medicine that consisted of 10 samples in each control and coronary artery disease group, with half of them biological males and the other half biological females in the age group of 60–70 and Non-Hispanic/Non-Latino White ethnicity and were non-smoker or former

smoker. The pooled plasma samples were also obtained commercially (Innovative Research # IPLAK2E10ML).

### Numerical simulations

Simulations were carried in COMSOL using 'Transport of Diluted Species' module to solve Nernst-Plank equation with irreversible reaction on the membrane surface located at the center of a 2D microfluidic channel. The two ends of the microfluidic channel were given open boundary conditions.

### Softwares used

All the analysis was performed on Graphpad Prism, ImageJ 1.53k, and MATLAB R2020b. Plots were created using Graphpad Prism and Adobe Illustrator while schematics were created using Autodesk 3ds Max, biorender.com and Adobe Illustrator. Proprietary software to Gamry Potentiostat was used for CVC curve acquisition. COMSOL was used for numerical simulations. Topspin was used for NMR acquisition.

### Reporting summary

Further information on research design is available in the Nature Portfolio Reporting Summary linked to this article.

## Data availability

Source data are provided with this paper. All the data supporting the findings of this study are available in the article, the supplementary information and from the corresponding authors upon request. Source data are provided with this paper.

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

## Acknowledgements

This work was partially supported by the NIH Commons Fund, through the Office of Strategic Coordination/Office of the NIH Director, 1UH3CA241684-01 (H.C.C. & S.S.), National Heart, Lung, and Blood Institute (NHLBI) under award number R01HL141909 (H.C.C. & S.S.). We would also like to acknowledge the core facilities at University of Notre Dame – confocal microscopy at Notre Dame Integrated Imaging Facility (Nikon C2+ Laser Scanning Confocal Microscope), NMR at NMR core facility (500 Hz Bruker Manual), Density Gradient Ultracentrifugation at Biophysics Core (Optima XPN-90), and zeta potentials were measured at CEST core (Nanobrook Omni). We would like to thank Dr. Jennifer Szymanowski and Actinide Research Facility at Notre Dame for allowing us access to Density Meter (Anton-Paar) and Dr. Sara Cole at NDIIF, Dr. Giselle Jacobson at BICF, and Dr. Evgenii Kovrigin at NMR core for helpful discussions and training at core instruments.

## Author contributions

S.K., S.S., and H.C.C. conceived the idea. S.S and H.C.C. led and organized the project. S.K. and S.S. designed the experiments and fabricated the microfluidic chip. S.K. and C.W. did ELISA microwell functionalization and blocking. S.K. conducted all experiments, including clinical samples in blind and developed the figures for the manuscript. S.K. and N.M. fabricated the sensors, and membrane functionalization was carried by S.K. All authors contributed to writing the manuscript.

## Competing interests

Authors declare no conflict of interests.
