## [Peer Review File · Nature Communications]

Quantifying PON1 on HDL with Nanoparticle-Gated Electrokinetic Membrane Sensor for Accurate Cardiovascular Risk AssessmentREVIEWER COMMENTS

Reviewer #1 (Remarks to the Author):

The authors presented here an interesting application of a method that utilizes an ion-exchange nanomembrane sensor, developed earlier. A crucial novelty lies here with the introduction of silica nanoparticles as a means to enhance the ion-current gating effect. The method is then applied to detect and quantify HDL-PON1 and total HDL as a marker for cardiovascular risk assessment. The prospect of the method for the assessment of cardiovascular risk is demonstrated by various means which include the analysis of patient samples as well as comparison with the other established methods.

However, there are some questions/concerns that require additional investigation/experiment. I, therefore, recommend a major revision of the manuscript addressing the concerns listed below

1. The use of silica nanoparticle for the amplification of charge seems to be the main novelty as far as the method is concerned. Yet, no experimental evidence has been presented to justify the selection of such particle or their size. Is it the optimum choice for such charge amplification? To what extent, the use of silica particle improves the detection sensitivity/LOD? Would that be the same with a smaller or larger particle size?
2. "This feature removes the need to conduct individual-sample calibration and a universal standard curve can be used for all samples, including plasma and serum." It is apparent that the size/charge of a target/probe will influence the electroconvective instability and therefore the signal from the sensor. In that case, it is unclear how a universal standard curve can be used as a calibration for any sample, since the charge, size aspect of different target will be different.
3. Use of silica nanoparticle is quite interesting but it brings several questions;
 - i) "The particle thus brings the necessary negative charge for the sensing signal..." With the functionalized surface and antibody coating on these particles, the surface charge of silica particles is expected to be very different and likely to be governed by the zeta potential of those antibodies. Different antibodies (if different in their zeta potential/isoelectric point) should produce different amount of charges. Authors should thoroughly investigate the influence arising from such effects, particularly in the case of multistep measurements using differently functionalized Si-NP. In this context, the use of oligonucleotides conjugated antibody may be a better alternative.
 - ii) With a much larger size (and charge) than the target, these particles are also expected to experience a much stronger steric hindrance, thereby, increasing the possibility that only a fraction of those bound target will be able to take part in the sandwich type interaction. This fraction will again depend on the surface concentration (and the bulk concentration) of the target. The saturation behavior observed may be an outcome of such a steric effect. However, in contrast to a one-step reporter addition, signal obtained with a multistep reporter addition will be dependent on the surface coverage of Si-NP of the previous step. It is therefore essential to identify the maximum target concentration in multistep measurement that can be reliably measured in such sandwich type assay.
4. The detection time has been claimed to be 20 mins which is only the probing time with the antibody coated silica NP. However, the major challenge, when it comes to rapid detection, arises from the capturing step of the target. Normally, as the concentration drops, it takes longer to achieve the equilibrium surface concentration due to diffusion limited transport. It is unclear how a "saturation" was obtained in 20 mins of sample injection when the concentration was varied by 3-4 orders of magnitude. Concentration dependent real-time measurement of the target binding may be done to get the required time for achieving an equilibrium surface concentration.
5. Since the Si-NP is the main driver for the observed signal, I wonder why there is a factor of 10 difference in saturation concentrations between PON1-HDL and PON1-free HDL in figure 2 (i-j)?
6. The control experiments presented in figure 4 shows that the Si-NP does not produce significant shift when the HDL is delipidated. While it proves the need to have the target for a signal, it does not necessarily show the extent of the signal that may be obtained due to non-specific interaction of the target with the Si-NP. This should be investigated by using isotype control antibody.
7. A minor mistake, Fig. 5 has been referred to as fig 4 in the text.
8. The estimation of ApoAI per HDL as presented in Fig 5f is also unclear. Furthermore, if there is a

distribution of number of ApoAI/HDL in the sample, additional correction might be needed. This is because the capturing of ApoAI-HDL will be positively biased towards the population having higher number of ApoAI/HDL.

Reviewer #2 (Remarks to the Author):

PON1-HDL has been proposed as a potential powerful risk marker for coronary heart diseases and other atherosclerotic diseases and expected to clear complex findings with HDL markers for cardiovascular risk, as it has been confused by failures in clinical trials targeting raising HDL and by epidemiological findings of risk increase in very-high-HDL populations. This parameter however is still to be established as a reliable risk marker by more specifically designed studies in both cross sectional and prospective, in order to identify whether this is a risk for preventive intervention, a marker to identify latent diseases or both. The authors developed a novel enzyme-free membrane sensor platform for quantification of PON1-HDL and total HDL, which they claim rapid and sensitive. The data presented seem solid and reliable, and the technique would provide a potential powerful tool to establish this parameter as an additional risk marker for cardiovascular diseases. They presented some cross-sectional data for identification of PON1-HDL as a cross-sectional “marker” of coronary heart disease by using their technique in a small population pilot study. The results are somewhat more powerful than other available measurement method for PON1-HDL and total PON1. The advantage of this technique should therefore be verified in more specifically designed full-scale population studies. The technique seems excellent but is not fully presented for its advantage in practical use in clinical risk finding.

Reviewer #3 (Remarks to the Author):

The article by Chang et al. presents a new method for the determination of PON1 associated with HDL based on a biosensor and suggests that it may be a good biomarker for cardiovascular disease based on the analysis of 10 patient samples and 10 controls. The article is well done and the potential applications of the new method are interesting, but in my opinion it is more appropriate for a specialized biosensor journal. Perhaps when the authors validate their results in a large series of patients, it can be considered for publication in Nature Communications or another generalist journal.

Reviewer #1 (Remarks to the Author):

The authors presented here an interesting application of a method that utilizes an ion-exchange nanomembrane sensor, developed earlier. A crucial novelty lies here with the introduction of silica nanoparticles as a means to enhance the ion-current gating effect. The method is then applied to detect and quantify HDL-PON1 and total HDL as a marker for cardiovascular risk assessment. The prospect of the method for the assessment of cardiovascular risk is demonstrated by various means which include the analysis of patient samples as well as comparison with the other established methods.

However, there are some questions/concerns that require additional investigation/experiment. I, therefore, recommend a major revision of the manuscript addressing the concerns listed below

We are glad that the reviewer finds our work interesting and novel, particularly the silica nanoparticle signal amplification technology. We have implemented major revision and believe we have address all his/her concerns.

1. The use of silica nanoparticle for the amplification of charge seems to be the main novelty as far as the method is concerned. Yet, no experimental evidence has been presented to justify the selection of such particle or their size. Is it the optimum choice for such charge amplification? To what extent, the use of silica particle improves the detection sensitivity/LOD? Would that be the same with a smaller or larger particle size?

We have incorporated a new paragraph on page 5 to explain our selection of the silica nanoreporter size. In our HDL quantification technique, the number of silica nanoparticle reporters must be the same as the target HDL. There is hence a lower bound to the particle size, roughly equal to the size of HDL at 15 nm. However, the more important cutoff size is the upperbound, as we would like to produce the largest signal possible. This upper bound is determined by Debye screening—charge on the particle that is more than one Debye length from the membrane surface would not produce a signal. The Debye length in our ion deplete region is 100nm. Hence, the upper bound of the particle size is ~ 65 nm (100 nm – 2 x 10 nm IgG – 15 nm HDL), which is roughly the size of our nanoreporter.

2. “This feature removes the need to conduct individual-sample calibration and a universal standard curve can be used for all samples, including plasma and serum.” It is apparent that the size/charge of a target/probe will influence the electroconvective instability and therefore the signal from the sensor.

In that case, it is unclear how a universal standard curve can be used as a calibration for any sample, since the charge, size aspect of different target will be different.

It is important to note that the signal is produced by the silica reporters and not by the captured HDL due to HDL's low surface charge. Therefore, even though each sample may have HDL of slightly different charge, it does not impact the signal. It is, however, quite possible that the coating of reporter probes on the silica can affect the surface charge density thus we have modified the statement on page 3 to – “This feature removes the need to conduct individual-sample calibration and a universal standard curve can be used for all samples, including plasma and serum for a given set of capture and reporter probes.”

We have also demonstrated that we do not need individual sample calibration for a given set of capture and reporter probes. We benchmarked PON1-HDL quantification by our NGEMS against a PON1-HDL assay with ELISA1 (a novel in-house ELISA that eliminates redox interference by soluble PON1--the scheme is shown in Extended Data Fig. 2a of the revised manuscript). Although these two quantitative assays measure PON1-HDL differently (electrokinetically vs redox reaction), we obtain Figure A below with high degree of correlation demonstrating we do not need individual sample calibration, as the NGEMS data are obtained from the same standard curve. We have inserted Fig. A as Fig. 6(a) in the revised paper.

Figure A: Comparison of PON1-HDL from NGEMS and ELISA1 showing calibration is unnecessary for every experiment for NGEMS

3. Use of silica nanoparticle is quite interesting but it brings several questions;

i) “The particle thus brings the necessary negative charge for the sensing signal...” With the functionalized surface and antibody coating on these particles, the surface charge of silica particles is expected to be very different and likely to be governed by the zeta potential of those antibodies. Different antibodies (if different in their zeta potential/isoelectric point) should produce different amount of charges. Authors should thoroughly investigate the influence arising from such effects, particularly in the case of multistep measurements using differently functionalized Si-NP. In this context, the use of oligonucleotides conjugated antibody may be a better alternative.

We agree with the reviewer that oligonucleotide could also be a potential candidate. As we mentioned in the response to the previous question, however, it is important that we have one reporter per HDL and have optimized the size of the silica nanoparticle with that consideration. It would be more difficult to tune the size of oligonucleotides, as the persistence length of a DNA is ionic strength dependent and considerable coiling can occur.

It is true that the surface charge of the silica particle can be altered by different antibodies functionalized to them. To quantify this sensitivity, we have calibrated the individual signal with each type of silica with their respective target of known concentration to establish a one-to-one correspondence between the voltage and bulk concentration of the target. Different antibodies for different targets do produce a slightly different calibration curves as shown in Fig. B (~ 40 mV variation in the linear range). Fig. B is now added as Fig. 2 on page 8 of the manuscript. The revision in response to comment 2 already states that calibration needs to be done for each reporter antibody. Nevertheless, we added to page 5 of the manuscript the following sentences: “The calibration curves allow us to establish one-to-one correspondence between voltage and bulk concentration, and also serve as a way to account for differences in silica surface charge caused by the different reporter probes on silica.”

Figure B: Different calibration curves for different species

(ii) With a much larger size (and charge) than the target, these particles are also expected to experience a much stronger steric hindrance, thereby, increasing the possibility that only a fraction of those bound target will be able to take part in the sandwich type interaction. This fraction will again depend on the surface concentration (and the bulk concentration) of the target. The saturation behavior observed may be an outcome of such a steric effect. However, in contrast to a one-step reporter addition, signal obtained with a multistep reporter addition will be dependent on the surface coverage of Si-NP of the previous step. It is therefore essential to identify the maximum target concentration in multistep measurement that can be reliably measured in such sandwich type assay.

For all our measurements, we operate in the dynamic range which is about an order of magnitude lower than the saturation concentration, and thus we expect the spacing between captured target to be large enough to accommodate silica reporter on all of the captured HDL. We also use high concentration of silica reporters ($\sim 30\text{nM}$) to enhance the binding probability. The linear range of our calibration curve should hence be free of any hindrance effects. Near saturation, however, steric effect can dominate. Also, earlier steps in a multi-step experiment can reduce the capacitance of the sensor and hence enhance the probability of saturation and steric hindrance. Fortunately, for this paper, we use only a single-step assay for our target PON1-HDL. Nevertheless, we have added these sentences to page 5: “Steric hindrance by the finite-size silica nanoparticles can contribute to the saturation of the signal, particularly for a multi-step titration assay (PON1-free HDL). It is hence important to operate away from saturation. All our experiments are done at least ten times lower than the saturation concentration.”

4. The detection time has been claimed to be 20 mins which is only the probing time with the antibody coated silica NP. However, the major challenge, when it comes to rapid detection, arises from the capturing step of the target. Normally, as the concentration drops, it takes longer to achieve the equilibrium surface concentration due to diffusion limited transport. It is unclear how a “saturation” was obtained in 20 mins of sample injection when the concentration was varied by 3-4 orders of magnitude. Concentration dependent real-time measurement of the target binding may be done to get the required time for achieving an equilibrium surface concentration.

We thank the reviewer for this extremely interesting and important question, and we have dedicated one figure and one extra page in the revised manuscript to address this issue. Our probing times of 20 mins and 60 mins for one concentration previously was for both the sample and silica incubation times (total 40 mins and 120 mins respectively). We have collected additional data at twenty minutes and one hour incubation times at the lower and higher end of the dynamic range which show that 20 minutes is enough to reach a steady state signal (pseudo steady state), as shown in Figure C(a-d) below. The reason we require only 20 minutes to reach a steady signal is because, at the high on-rate of the antigen-antibody docking, we operate in the mass-transfer limited regime. For a small sensor of size a ($\sim 100\mu\text{m}$) operating in the in the mass-transfer limited region, rapid depletion of analyte above the membrane occurs within a depletion volume of linear dimension a . This depletion occurs within a time of a^2/D , roughly 4 - 7 minutes for HDL, where D is the diffusivity of the particle. After this time, the flux is significantly lower as the system asymptotically reaches the true reversible adsorption equilibrium over a long time. We have conducted new experiments to verify that we are operating in this mass-transfer limited region. We changed the surface antibody concentration to one-twentieth of its original value and found the voltage shift to remain unchanged with the same bulk analyte concentration (new Figure Ce). Since the capture rate depends on the surface concentration of the antibody only in the kinetically limited regime, we have hence shown that we are in the mass-transfer controlling regime. Operating in this regime requires detection of very small number of reporters which can be achieved with our platform but not with ELISA or other commonly used platform due to their 10 million to 1 billion reporter requirement to detect the signal. A good discussion of how the size of a sensor can favor the diffusion-limited region is offered by Lemay and Moazzenzade, *Analytical Chem*, 93, 9023(2021), which is now added as reference 49.

We have also done some numerical simulation of this rapid depletion (Figure Cf-i). As can be seen, majority of the depletion occurs within the first few minutes, and the surface concentration of the analyte on the membrane does not change over time significantly. The concentration profile directly above the membrane and at the center of the channel is also shown. From our work on how surface

charge affects the voltage shift (Ref 42 in the manuscript), the voltage signal is $\frac{2RT}{F} \ln \left(\frac{z_s C_s}{c_0 \lambda} \right)$ where C_s is the surface concentration of the analyte on the surface, z_s is the charge on silica nanoreporters, c_0 is ionic concentration of water ($10^{-7}M$) and $\lambda = 100nm$ is the debye layer at DI conditions. As the voltage signal has a logarithmic dependence on the surface concentration (ref 42), the voltage signal does not increase significantly after a^2/D to reach a pseudo-steady state (prior to a very slow diffusive flux towards true equilibrium, see Figure Cj below). We have created a new subsection on page 6 in the manuscript, which includes Figure C below as **Figure 4** in the manuscript, to address this rapid incubation time issue. This observation really elevates the significance of our sensor and this manuscript. We thank the reviewer for pointing it out.

Figure C: (a)-(d) Impact of incubation time on voltage shift for different PON1-HDL and total HDL measurements. (e) Effect of surface coverage on antibodies on the voltage signal. Numerical simulation results showing (f) variation of concentration at channel center (membrane at $x/w, y/h=0$), (g) variation of concentration vertically above the membrane, (h) Surface concentration of the analyte and the dimensionless flux at different times, (i) zoomed-in channel at multiple times, and (j) change in signal with time.

5. Since the Si-NP is the main driver for the observed signal, I wonder why there is a factor of 10 difference in saturation concentrations between PON1-HDL and PON1-free HDL in figure 2 (i-j)?

We believe this is because of the nature of the sample used during the calibration. We used isolated HDL which contained only ~5-10% PON1-HDL. At saturation for total HDL, the capture antibody is saturated by HDL, out of which only 5-10% will be PON1-HDL. Thus only about one-tenth of the expected silica particle at the saturation compared to its total HDL (HDL-P) counterpart. If we could isolate pure PON1-HDL in its native form without other HDL, it is quite possible that saturation signal would be identical.

6. The control experiments presented in figure 4 shows that the Si-NP does not produce significant shift when the HDL is delipidated. While it proves the need to have the target for a signal, it does not necessarily show the extent of the signal that may be obtained due to non-specific interaction of the target with the Si-NP. This should be investigated by using isotype control antibody.

Our previous data in Manuscript-Figure 4 showed this antibody control as anti-ApoB which should not bind to any HDL. However, we agree with the reviewer that an isotype control would also be a more appropriate choice. Therefore, we have performed the experiment with an isotype control in the manuscript as seen in Figure Ce (new Figure 4e in the manuscript). We can hence rule out non-specific binding.

7. A minor mistake, Fig. 5 has been referred to as fig 4 in the text.

This mistake is now corrected.

8. The estimation of ApoAI per HDL as presented in Fig 5f is also unclear. Furthermore, if there is a distribution of number of ApoAI/HDL in the sample, additional correction might be needed. This is because the capturing of ApoAI-HDL will be positively biased towards the population having higher number of ApoAI/HDL.

Figure D: ApoAI vs HDL-P gives a slope of about 5 with human plasma samples.

(For clarity, we reproduce the figure in Figure D above. It is now Figure 6f in the revised manuscript.) We thank the reviewer for this question and we have added more explanation in the revision. For detection of HDL, an important metric that is commonly used in the literature is to plot the total HDL particle concentration against the total ApoAI concentration from actual human patient samples. The methods that produce a slope of 3-5 are considered more accurate because they predict 3-5 ApoAI per HDL consistent with the known structure of HDL. Our platform does that but other methods such as NMR are known to produce <1 ApoAI per HDL or Ion Mobility Analyzer for HDL which produces >10 ApoAI per HDL. Therefore, the figure's purpose is to show our platform shows an accurate slope as expected for the correct measurements. Also, because we are in mass transfer limited regime where depletion occurs, we do not have a bias in the capture step (variation in k_{on} or affinity). There could be a bias due to different HDL diffusivity with different ApoAI number, but that should be negligible. Additionally, the reporter negates any bias by only binding to the target on the surface. This is seen in the PON1-HDL from NGEMS and ELISA1 as they are consistent. We have added the following lines to our manuscript on page 13 –

“Additionally, we use the unique structural property that the HDL contain 2-8 ApoAI per HDL to evaluate our platform by cross-plotting total ApoAI against HDL-P from our platform in Fig. 6f, which is a standard evaluation test for HDL platforms. A slope of 3-5 is expected if the platform counts HDL-P correctly. “

Reviewer #2 (Remarks to the Author):

PON1-HDL has been proposed as a potential powerful risk marker for coronary heart diseases and other atherosclerotic diseases and expected to clear complex findings with HDL markers for cardiovascular risk, as it has been confused by failures in clinical trials targeting raising HDL and by epidemiological findings of risk increase in very-high-HDL populations. This parameter however is still to be established as a reliable risk marker by more specifically designed studies in both cross sectional and prospective, in order to identify whether this is a risk for preventive intervention, a marker to identify latent diseases or both. The authors developed a novel enzyme-free membrane sensor platform for quantification of PON1-HDL and total HDL, which they claim rapid and sensitive. The data presented seem solid and reliable, and the technique would provide a potential powerful tool to establish this parameter as an additional risk marker for cardiovascular diseases. They presented some cross-sectional data for identification of PON1-HDL as a cross-sectional “marker” of coronary heart disease by using their technique in a small population pilot study. The results are somewhat more powerful than other available measurement method for PON1-HDL and total PON1.

We thank the reviewer for his/her insights on the need for better risk marker, especially the observations on failed clinical trials. It is also gratifying to us that the reviewer, who is obviously an expert in the field, found our work novel. Additionally, we have not come across any other PON1-HDL study and its effect on cardiovascular risk evaluated directly for human patients. We hence believe that ours is the first study to do so and we have tried to highlight the significance of this work in the abstract.

The advantage of this technique should therefore be verified in more specifically designed full-scale population studies. The technique seems excellent but is not fully presented for its advantage in practical use in clinical risk finding.

We now discuss in detail the automation of our platform and our effort to advance the technology to full-scale population studies. Because our platform is microfluidics-based, it can be fully automated with peristaltic pumps and potentiostats/galvanostats. A voltage or current sweep across the IV curve which is <4V and <200 microamperes means that the power requirement of such a setup can be supplied with the help of a small power supply for both the pumps and the potentiostat. The setup that we are currently developing eliminates the need for trained technicians to perform the test since

the whole experiment will be fully automated. One of the first versions of the prototype is shown in Figure E and we plan to build a second-generation prototype soon for large-scale studies in the field to demonstrate the usefulness of our platform. Turn-key operation, low cost and fast measurement is the end-goal in terms of automation of this platform. Additionally, our platform can, in theory, be extended to identify any other proteomic subclass of HDL in their native form which cannot be done by any of the commonly used techniques, often relying on surrogate measurement of the interested subclass.

The following paragraph is appended to the manuscript on page 15 :

“Due to its microfluidics nature, our platform can be fully automated with peristaltic pumps and potentiostats/galvanostats. A voltage or current sweep across the IV curve which is $<4V$ and <200 microamperes means that the power requirement of such a setup can be supplied with the help of a small power supply for both the pumps and the potentiostat. A first generation prototype is currently in development. Turn-key operation, low cost and fast measurement is the end-goal in terms of automation of this platform. After receiving feedback from clinicians for the first prototype, we plan to develop a second-generation prototype to conduct a large-scale clinical study that would demonstrate the usefulness of our platform and the validity of PON1-HDL as a marker. Our platform can also, in theory, identify any other proteomic subclass of HDL in their native form, which cannot be done by any of the commonly used techniques. They often relying on surrogate measurement of the interested subclass⁵⁸. “

Figure E: One of the first fully automated versions of the prototype developed.

Reviewer #3 (Remarks to the Author):

The article by Chang et al. presents a new method for the determination of PON1 associated with HDL based on a biosensor and suggests that it may be a good biomarker for cardiovascular disease based on the analysis of 10 patient samples and 10 controls. The article is well done and the potential applications of the new method are interesting, but in my opinion it is more appropriate for a specialized biosensor journal. Perhaps when the authors validate their results in a large series of patients, it can be considered for publication in Nature Communications or another generalist journal.

We are glad the reviewer found our work well-done and interesting. We also agree with his/her opinion on validating the study with a larger group of patients. We are currently raising funds and recruiting clinical partners to conduct such a large study. We currently do have a fully automated prototype of our setup that can perform the experiments with relatively minimal training but requires further work to improve and should be ready in the near future.

However, we disagree with the reviewer that this paper should not be published in Nature Communication. All reviewers found our work novel and the first two reviewers also found it significant. Despite our small cohort, our current pilot test with PON1-HDL achieved an AUC of 0.944-1.036, which outperforms AUC of every other biomarker currently reported in the literature. Moreover, this is the first study to compare PON1-HDL from healthy and patient samples of cardiovascular disease instead of total HDL or total PON1. This is only possible because our new sensor has sufficient sensitivity for the PON1-HDL assay, as reviewer 1 highlighted. Our study shows that total PON1 weakly correlated with CVD risk compared to PON1-HDL. Moreover, we have compared the AUC values of the other biomarkers tested (on independent platforms) and they had similar AUC values as reported in the literature (shown in Figure F). These are significant results that would be appreciated by the general scientific community, including both the CDC biomarker community and the sensor community.

Figure F: AUC of commonly used biomarker for our current pool of patients and their values from other independent study.

REVIEWERS' COMMENTS

Reviewer #1 (Remarks to the Author):

The authors have addressed all the concerns/suggestions with additional data. The revised version may now be accepted.

Reviewer #2 (Remarks to the Author):

This reviewer has no doubt on novelty and soundness of the technique for the method the authors report. PON1-HDL is a potential and likely-specific negative risk marker for CHD. However this marker has not yet been established as a clinically useful tool to predict any risk of the patients. Therefore, scientific merit of this work is limited to technical development for an unestablished risk marker, and it would be powerful and useful in order to identify and establish if this marker is really useful for risk finding in the future. The clinical data presented in the manuscript indicates that low level of PON1-HDL measured by the method reflects the presence of CHD in the patients but does not predict their future risk.

Reviewer #3 (Remarks to the Author):

The authors have clarified to me the aspects in which I had doubts about their manuscript and, therefore, I consider it acceptable for publication in Nat. Comm.

Reviewer #1 (Remarks to the Author):

The authors have addressed all the concerns/suggestions with additional data. The revised version may now be accepted.

We thank the reviewer for recommending acceptance of our manuscript.

Reviewer #2 (Remarks to the Author):

This reviewer has no doubt on novelty and soundness of the technique for the method the authors report. PON1-HDL is a potential and likely-specific negative risk marker for CHD. However this marker has not yet been established as a clinically useful tool to predict any risk of the patients. Therefore, scientific merit of this work is limited to technical development for an unestablished risk marker, and it would be powerful and useful in order to identify and establish if this marker is really useful for risk finding in the future. The clinical data presented in the manuscript indicates that low level of PON1-HDL measured by the method reflects the presence of CHD in the patients but does not predict their future risk.

We thank the reviewer for their suggestion and feedback. We agree that a larger patient cohort needs to be tested and have incorporated this into our abstract.

Reviewer #3 (Remarks to the Author):

The authors have clarified to me the aspects in which I had doubts about their manuscript and, therefore, I consider it acceptable for publication in Nat. Comm.

We thank the reviewer for recommending acceptance of our manuscript.